# Determining a Threshold to Delimit the Amazonian Forests from the Tree Canopy Cover 2000 GFC Data

**DOI:** 10.3390/s19225020

**Published:** 2019-11-18

**Authors:** Kaio Allan Cruz Gasparini, Celso Henrique Leite Silva Junior, Yosio Edemir Shimabukuro, Egidio Arai, Luiz Eduardo Oliveira Cruz e Aragão, Carlos Alberto Silva, Peter L. Marshall

**Affiliations:** 1Divisão de Sensoriamento Remoto, Instituto Nacional de Pesquisas Espaciais, São José dos Campos – SP, Brazil; celso.junior@inpe.br (C.H.L.S.J.); egidio.arai@inpe.br (E.A.);; 2Department of Geographical Sciences, University of Maryland, College Park, Maryland, MD 20740, USA; carlos_engflorestal@outlook.com; 3Department of Forest Resources Management, The University of British Columbia, 2424 Main Mall, Vancouver, BC V6T 1Z4, Canada; peter.marshall@ubc.ca

**Keywords:** forest mapping, Google Earth Engine, REDD+, remote sensing, forest degradation

## Abstract

Open global forest cover data can be a critical component for Reducing Emissions from Deforestation and Forest Degradation (REDD+) policies. In this work, we determine the best threshold, compatible with the official Brazilian dataset, for establishing a forest mask cover within the Amazon basin for the year 2000 using the Tree Canopy Cover 2000 GFC product. We compared forest cover maps produced using several thresholds (10%, 30%, 50%, 80%, 85%, 90%, and 95%) with a forest cover map for the same year from the Brazilian Amazon Deforestation Monitoring Project (PRODES) data, produced by the National Institute for Space Research (INPE). We also compared the forest cover classifications indicated by each of these maps to 2550 independently assessed Landsat pixels for the year 2000, providing an accuracy assessment for each of these map products. We found that thresholds of 80% and 85% best matched with the PRODES data. Consequently, we recommend using an 80% threshold for the Tree Canopy Cover 2000 data for assessing forest cover in the Amazon basin.

## 1. Introduction

Tropical forests play an essential role in the carbon (C) cycle [1]. However, human actions through deforestation and forest degradation can revert the natural tropical forest C sink. It is estimated that deforestation is the second largest source of C emission into the atmosphere, emitting between 0.81 and 1.14 Pg C annually [2,3,4,5]. Degradation is another important driver of carbon emission within tropical regions. In the Amazon, during drought years, C emissions from forest fires can exacerbate old-growth forest deforestation emissions by more than half [6].

Brazil has been monitoring the extent of the Amazon rainforest since the late 1980s using Landsat images available through the Brazilian Amazon Deforestation Monitoring Project (PRODES), a project operated by the Brazilian National Institute of Space Research in collaboration with the Ministry of the Environment and the Brazilian Institute of Environment and Natural Resources [7]. Since deforestation is the second largest CO_2_ emission driver [8], PRODES allowed Brazil to be a pioneer in reducing emissions under the Reducing Emissions from Deforestation and Forest Degradation (REDD+) policy [9]. For the correct monitoring of forest areas, there is a need to draw a baseline, or year of reference, to begin monitoring. Consequently, a map representing the forest area at the beginning of forest monitoring is important [10]. With the baseline forest (a forest mask) established, one can quantify new deforestation as well as degraded areas by identifying the affected areas on the initial map. Recently, such forest monitoring initiatives have gained popularity [11,12], especially after the emergence of remote sensing applications using cloud computing as a platform [13].

The Google Earth Engine platform has allowed studies on a planetary scale [14]. To date, the uses of the Google Earth Engine have been diverse, such as mapping malaria risks [15], environmental monitoring [13,16], and urban mapping [17,18]. Hansen et al. [13] were among the first researchers to use the Google Earth Engine platform. They mapped forest changes worldwide from 2000 to 2013 using Landsat imagery collection (30 m of spatial resolution). In the current version of the mapping engine (v1.6), the authors processed images up to 2018 [19]. A highlight of their research was the construction of a raster map of the 2000 baseline named ‘Tree Canopy Cover 2000 GFC Data’ (TreeCover2000). TreeCover2000 represents tree cover for the year 2000 at a 30-meter spatial resolution. These data were generated using multispectral satellite imagery from the Landsat 7 thematic mapper plus (ETM+) sensor [13,20]. Cloud-free observations from over 600,000 images were analyzed using Google Earth Engine to determine per-pixel tree cover using a supervised machine learning algorithm (regression tree approach) [13,20]. The regression tree training data were obtained from Ikonos (4 m of spatial resolution) and QuickBird (2.8 m of spatial resolution) image classifications, in which tree cover was transformed to percent values [21,22]. The result was a map of percentage values of tree cover, ranging from 0% to 100%, where 0% represented no tree cover and 100% represented maximum tree cover for a pixel. These data require a user to choose a percentage threshold value to determine whether a pixel is considered forest (i.e., cover value equal to or greater than the threshold). Several thresholds have been empirically chosen in the literature to define forest cover with the TreeCover2000 data. For example: Grecchi et al. [23], Taubert et al. [24], Brinck et al. [25], and Esquivel-Muelbert et al. [26] used 30%; Shimabukuro et al. [27] used 50%; Wagner et al. [28] used 80%; and Tyukavina et al. [29] used 83%. However, as far as we know, no study has focused on testing the performance of these thresholds in delimiting the Amazonian rainforests in order to match with other countries’ definition of forest. Without a threshold standardization, comparisons among studies and with official statistics become impossible.

Bearing in mind that the Amazonian forest extends beyond Brazil’s borders and, apart from Brazil, few other Amazonian countries have long-term, large-scale forest mapping projects in place, the use of the TreeCover2000 dataset is essential for establishing baselines of intact forest areas. It is also valuable to have measurements that are comparable across countries. However, it is not clear what threshold would best delimit Amazonian forests. To answer this question, we compared the PRODES forest cover map with maps produced using different thresholds of the TreeCover2000 data for an area within the Brazilian portion of the Amazon basin. We also compared the classifications in all the maps to 2550 independently assessed Landsat pixels indicating forest coverage in the year 2000.

## 2. Materials and Methods

### 2.1. Study Area

We used the Brazilian state of Mato Grosso (Figure 1) for this analysis due to its physiographic heterogeneity. Mato Grosso includes flooded fields (Pantanal), natural fields and savannas (Cerrado), and deciduous and evergreen forests (Amazonia) [30]. This region is also part of the “arc of deforestation”, an area of intense deforestation that borders the Cerrado [31] and recently considered a consolidated agricultural frontier [32]. Consequently, Mato Grosso is potentially the only place that provides the full range of phytophysiognomies found in the Amazon basin as a whole.

### 2.2. Datasets

The Tree Canopy Cover 2000 GFC data (TreeCover2000) were obtained from the Global Forest Change website [19]. These data represent the percentage of all vegetation greater than 5 m in height, not necessarily natural forest, with values ranging from 0% (no vegetation greater than 5 m in height) to 100% (total cover) [13]. The global dataset is divided into 10-degree by 10-degree segments (tiles). Four such tiles were necessary to get complete coverage of Mato Grosso (00N_060W, 00N_070W, 10S_60S, and 10S_70S). The spatial resolution of the data is 0.00025 degrees, which is equivalent to 30 m in the Equator region.

The PRODES data were obtained from [34]. The data have a 1:250,000 mapping scale and represent areas of old-growth forests mapped manually from radar and optical remote sensing data. We used the vector data from 2014. From that, we reconstructed the forest cover for the year 2000 by converting all deforested polygons from the period 2001 to 2014 back to forest, with the remainder labelled as non-forest. These data were then converted to a 30 m raster basis to be compatible with Tree Canopy Cover 2000 data.

An independent reference sample of forest and non-forest coverage in the study area was obtained from Tyukavina et al. [29,35]. In their study, 10,000 Landsat (5, 7, and 8) 30 m pixels were classified over the period from 1999 to 2013 across the entire Brazilian Amazon. Of these sample pixels, 2550 were located in Mato Grosso. As this work is based on the year 2000, an external evaluator assessed each of the 2550 pixels in Mato Grosso and labelled each as forest or non-forest based on the year 2000 data.

### 2.3. Methods

About 9329 10 by 10 km plots (samples) were established across Mato Grosso (Figure 1). Of these plots, 4994 (53%) were in the Amazon biome, 3703 (40%) were in the Cerrado biome, and 632 (7%) were in the Pantanal biome. Initially, we calculated the forest cover percentage from the PRODES data within each of the 9329 plots. Then, we calculated the forest cover proportion from the Tree Canopy Cover 2000 maps, which used the following thresholds: (1) greater than 10%; (2) greater than 30%; (3) greater than 50%; (4) greater than 80%; (5) greater than 85%; (6) greater than 90%; and (7) greater than 95%. Any pixel that exceeded these thresholds was considered as forest. We selected these thresholds based on preliminary tests and the previous studies from the literature [23,24,26,27,28,29], representing a wide range of thresholds. Hereafter, we refer to these thresholds as 10%, 30%, and so on.

The proportions of forest cover within each of the 9329 plots defined by each threshold and the PRODES forest cover were compared using linear regression, suggested by Shimabukuro et al. [30]. This method is a robust and straightforward approach to validating remote sensing mapping. The coefficient of determination (R^2^), intercept, slope, and the root mean squared error (RMSE) were used for comparison among the various regression equations to determine the best threshold to use. The ideal value for R^2^ was 1, 0 for the intercept, 1 for the slope, and 0 for the RMSE. We used the bootstrap method for this analysis. In this approach, 10,000 interactions were performed, where each interaction was randomly raffled 10% of the 9329 plots with replacement. Then, the means and standard deviations of the 10,000 R^2^, intercept, slope, and the RMSE were calculated. All analyses were performed using the R statistical software package [36]. To calculate the forest proportion within each of the 10 by 10 km plots, the “lsm_c_pland” function of the “landscapemetrics” package [37] was used. Linear regressions were performed using the "ln" native function of the R software.

Moreover, we built a confusion matrix that was used to tabulate the differences in classification between the maps and the reference samples using the “Accuracy Assessment” tool [38], developed by the Food and Agriculture Organization of the United Nations (FAO), from the 2550 Landsat sample pixels (year 2000) provided by Tyukavina et al. [29]. As suggested by Olofsson et al. [39] and the FAO [38], we determined a weighted matrix error using the ratio of the area of each class sampled over the total area of each class (Equation (1)). This weighted error was then used to calculate the standard error (Equation 2) and the confidence interval for the estimated area of each class,
(1)$${\hat{p}}_{i,j}=\frac{W_{i}\cdot n_{i,j}}{n_{i}}$$
(2)$$S\left(\hat{p}\right)=\sqrt{\underset{i}{\sum }\frac{W_{i}{\hat{\cdot p}}_{i,j}-{\hat{p}}_{i,j}^{2}}{n_{i}-1}}$$
where ${\hat{p}}_{i,j}$ is the proportional area for each cell in the matrix, $W_{i}$ corresponds to the class weights (the proportional area of class i), $n_{i,j}$ is the sample count in cell i,j, $n_{i}\;$ is the total sample count in class i, and $S\left(\hat{p}\right)$ is the standard error of the area estimates assuming simple random sampling.

## 3. Results

Our analyses revealed that the lower thresholds (10%, 30%, and 50%) included forest cover in areas covered by savanna vegetation within the Cerrado biome (see Figure 1) [40], in the south-central portion of Mato Grosso (see Figure 2 and Figure 3) [40]. Maps produced using the 80%, 85%, and 90% thresholds showed greater visual similarity to the PRODES forest map (Figure 2 and Figure 3).

As expected, the 95% threshold map indicated the smallest forest cover, about 37% of Mato Grosso, and the 10% threshold map indicated the largest forest cover, about 66% of Mato Grosso (Table 1). The PRODES map indicated approximately 41% of forest cover within Mato Grosso in 2000, which was most closely matched by the 90% threshold.

We found that forest cover maps produced from the 10%, 30%, and 50% thresholds tended to overestimate forest cover based on the 9329 sample plots, as evidenced by the concentration of the majority of the points above the 1:1 line (Figure 4). The maps produced using thresholds of 90% and 95% underestimated forest cover, as more points fell below the 1:1 line than above it. Finally, the forest maps produced using the 80% and 85% thresholds showed a more even dispersion of points around the regression line and the 1:1 line.

All the 10,000 regressions were significant at a level of 1% (*p*-values less than 0.001). The R^2^ increased from an average of 0.612 ± 0.021 to 0.869 ± 0.013 for 10% and 85%, respectively. The intercept coefficient values decreased from an average of 41.560 ± 1.058 for the 10% threshold to an average of 0.978 ± 0.473 for the 95% threshold. At the same time, the slope coefficient values increased from an average of 0.591 ± 0.013 for the 10% threshold to an average of 0.888 ± 0.009 for the 85% threshold. Finally, the RMSE had a lower average of 13.060 ± 0.644 for the 85% threshold, and a higher average of 17.910 ± 0.498 for the 10% threshold.

The accuracy assessment results showed that the poorest match (lowest overall accuracy) was obtained with the 95% threshold (71%) and the best match (highest overall accuracy) was obtained with the 50% threshold (87%) (Table 2). The PRODES data had the third lowest match (74%), albeit slightly higher than the 90% threshold. The 80% threshold best matched with the PRODES data in terms of the estimated cover of non-forest and forest in Mato Grosso.

Not surprisingly, the highest user’s accuracy (85%) for non-forest cover was obtained with the 10% threshold and declined with increasing threshold size (Table 3). The highest user’s accuracy (99%) for forest cover was found with the 95% threshold and declined with decreasing thresholds. The pattern for the producer’s accuracy was the opposite of that of the user’s accuracy. The highest producer’s accuracy for non-forest cover (99%) was found for the 95% threshold and the highest producer’s accuracy for forest cover (94%) was found for the 10% threshold. The thresholds that best resembled those obtained by the PRODES data were 85% (user’s accuracy) and 50% (producer’s accuracy).

## 4. Discussion and Conclusions

We compared Amazonian forest cover using Tree Canopy Cover 2000 and the PRODES data within a portion of the Amazon basin in Brazil. By creating several forest maps from applying thresholds to the Tree Canopy Cover 2000 data, we showed that there are relevant differences in the quantity and spatial distribution of forest cover depending on the threshold chosen. For instance, we found that thresholds of 80% and 85% for the Tree Canopy Cover 2000 data best matched the PRODES forest cover data across Mato Grosso State in Brazil. For example, the lower thresholds evaluated—between 10% and 50%—overestimated forest cover, especially within savanna formations, where trees are sparser within the landscape. Thus, the choice of less conservative thresholds may inflate the quantification of forest disturbances, such as fires [27], deforestation [13], selective logging [23], forest fragmentation [24,25,26], and, consequently, carbon emissions accounting.

We emphasize that the choice of a threshold level for the Tree Canopy Cover 2000 data will depend on the nature of the application. If the change in forest cover through time is being assessed, then, obviously, consistency through time among threshold levels is more important than the threshold level ultimately chosen for the assessment. However, in general, based on our analyses, we recommend using an 80% threshold with the Tree Canopy Cover 2000 data for assessing forest cover in the Amazon basin.

The approach we presented may help countries currently involved in REDD+ in the Amazon region. Forest reference levels can be based on Tree Canopy Cover 2000 data using our suggested threshold (UNFCCC, 2010) [41]. Presently, Brazil, Colombia, Ecuador, Guyana, Peru, and Suriname are members of REDD+ [42]. A standardization of methods across countries would be beneficial for the evaluation of the effectiveness of countries’ efforts to mitigate emissions from land cover change.

## Figures and Tables

**Figure 1 sensors-19-05020-f001:**
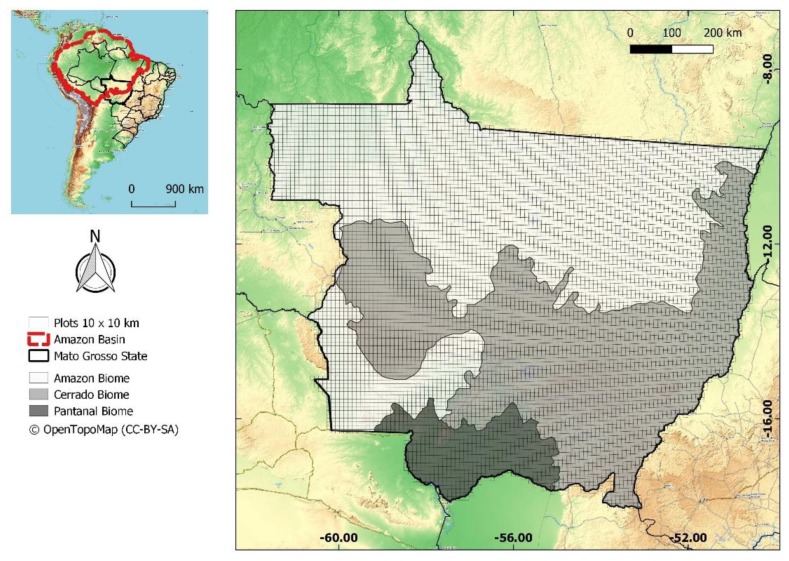
Location of the state of Mato Grosso, representative biomes, and the spatial distribution of the 10 km^2^ plots used as samples. The smaller map shows the extent of the Amazon basin [33].

**Figure 2 sensors-19-05020-f002:**
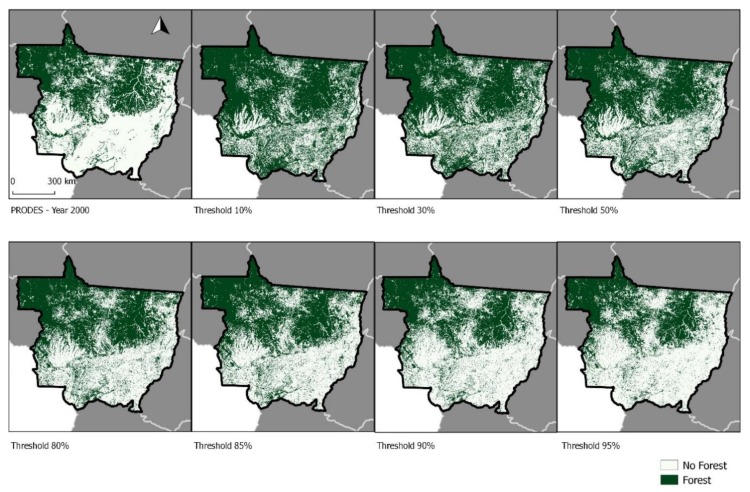
Spatial arrangement of each threshold assessed in the study compared to the Brazilian Amazon Deforestation Monitoring Project (PRODES) data.

**Figure 3 sensors-19-05020-f003:**
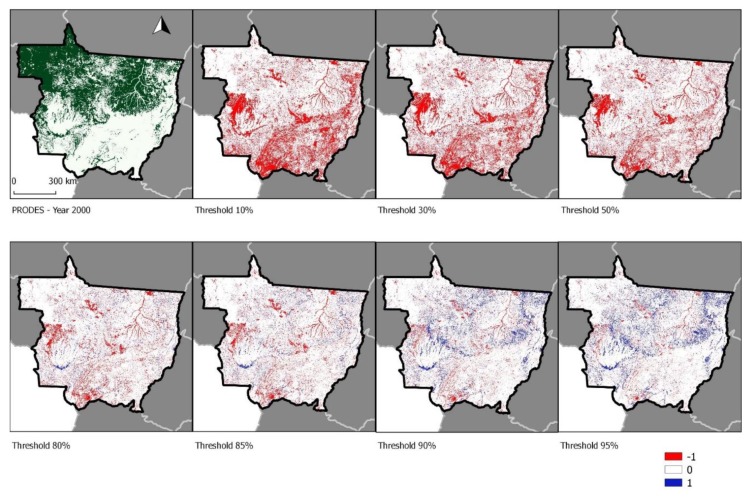
Maps of differences between spatial arrangements of each threshold assessed in the study compared to the PRODES data. Red (−1) represents pixels that were non-forest cover using PRODES and forest cover using the Tree Canopy Cover 2000 data. White (0) represents pixels classified into the same class using both datasets. Blue (1) represents pixels that were classified as forest cover using PRODES and non-forest cover using Tree Canopy Cover 2000 data.

**Figure 4 sensors-19-05020-f004:**
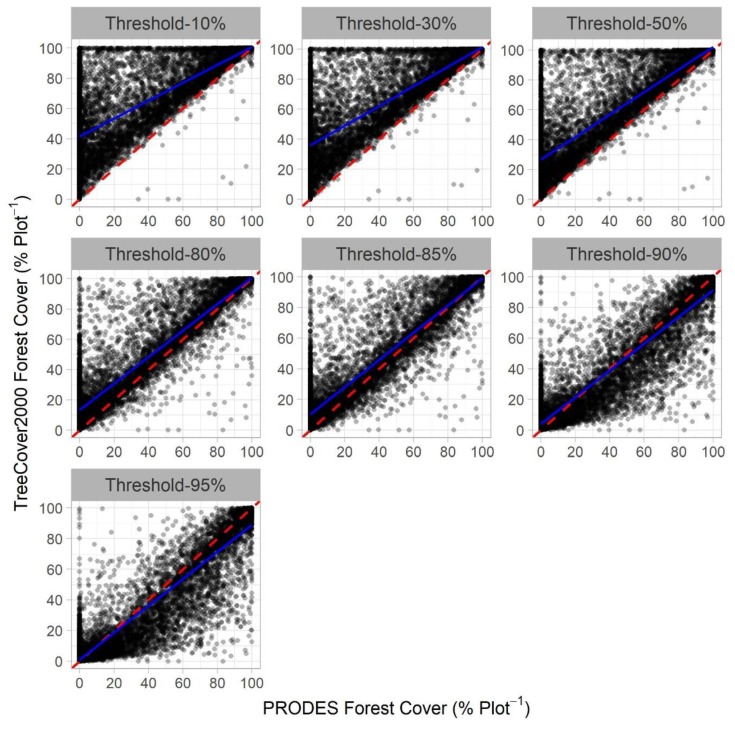
Regression between the forest percentage within all of 9329 (10 by 10 km) samples cells using different thresholds from the Tree Canopy Cover 2000 data and their corresponding percentages on a reference map developed using the PRODES data. The dashed red line is the 1:1 line. The blue line is the average regression line from 10,000 interactions for each threshold tested.

**Table 1 sensors-19-05020-t001:** Forest cover assessment for the year 2000 in Mato Grosso, Brazil, according to the PRODES map and maps produced using various thresholds with the Tree Canopy Cover 2000 data.

Map	Forest (%)	Non-Forest (%)
PRODES map	41	59
10% Threshold	66	34
30% Threshold	63	37
50% Threshold	58	42
80% Threshold	49	51
85% Threshold	47	53
90% Threshold	40	60
95% Threshold	37	63

**Table 2 sensors-19-05020-t002:** Result from 10,000 bootstrap interactions. Intercept, slope, coefficient of determination (R^2^), root mean squared error (RMSE), and *p*-values of the linear regressions are used for comparing the tested thresholds to the PRODES data. SD is the standard deviation.

Threshold	R^2^ (± SD)	Intercept (± SD)	Slope (± SD)	RMSE (± SD)	*p*-value (± SD)
10%	0.612 ± 0.021	41.560 ± 1.058	0.591 ± 0.013	17.910 ± 0.498	0 ± 0
30%	0.669 ± 0.020	35.870 ± 1.029	0.655 ± 0.013	17.520 ± 0.542	0 ± 0
50%	0.752 ± 0.019	26.630 ± 0.953	0.755 ± 0.012	16.500 ± 0.620	0 ± 0
80%	0.860 ± 0.014	13.290 ± 0.693	0.874 ± 0.009	13.380 ± 0.646	0 ± 0
85%	0.869 ± 0.013	10.351 ± 0.693	0.888 ± 0.009	13.060 ± 0.644	0 ± 0
90%	0.857 ± 0.013	4.120 ± 0.513	0.872 ± 0.010	13.540 ± 0.550	0 ± 0
95%	0.852 ± 0.013	0.978 ± 0.473	0.879 ± 0.010	13.960 ± 0.010	0 ± 0

**Table 3 sensors-19-05020-t003:** Error matrix, accuracy, and confidence interval for each class (forest and non-forest) at each threshold compared to the PRODES data.

Predictions	Class	Observed		Estimated Area (%)	Estimated Area ± 95% Confidence (km^2^)	User’s Accuracy (%)	Producer’s Accuracy (%)	Overall Accuracy (%)
		Non-Forest	Forest					
10%	Non-forest	544	96	39.3	352,159 ± 12,849	85	64	84
	Forest	300	1601	60.7	543,914 ± 12,849	84	94	
30%	Non-forest	600	118	39.4	353,006 ± 12,606	84	71	86
	Forest	244	1579	60.6	543,067 ± 12,606	87	93	
50%	Non-forest	688	186	38.6	346,277 ± 12,568	79	82	87
	Forest	156	1511	61.4	549,796 ± 12,568	91	74	
80%	Non-forest	787	389	36.0	322,227 ± 13,082	67	93	82
	Forest	57	1308	64.0	573,846 ± 13,082	96	77	
85%	Non-forest	803	461	35.2	315,401 ± 13,256	64	95	80
	Forest	41	1236	64.8	580,672 ± 13,256	97	73	
90%	Non-forest	821	651	34.3	307,281 ± 13,985	56	97	73
	Forest	23	1046	65.7	588,792 ± 13,985	98	62	
95%	Non-forest	832	737	33.6	302,454 ± 14,090	53	99	71
	Forest	12	960	66.4	593,618 ± 14,090	99	57	
PRODES	Non-forest	748	572	36.6	327,194 ± 15,134	57	89	74
	Forest	96	1125	63.4	567.323 ± 15,134	92	66	

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
