# Peer review of "Determining a Threshold to Delimit the Amazonian Forests from the Tree Canopy Cover 2000 GFC Data"

_sensors, 2019, doi:10.3390/s19225020_

Round 1

Reviewer 1 Report

The paper introduces a method to estimate tree cover in the Amazon basin based on thresholding the available Treecover2000 data. The analysis is basically the comparison of thresholded results to a public reference data set.

The method leads to a valid result; nevertheless, the paper lacks scientific aspects.

First, all data are previously available and the authors did not process them. Their work is a comparison by using thresholds between 0-100 to determine the best fit. In this step, the work is very naive and the tested values not justified. They just appear. Why not 82 instead of 80%?

It is not clear if the authors did a rigorous separation of the data set into test and control sets. It is recommended to use a set for finding the threshold and another one to verify if the method really works.

The contribution of the paper is weak because a similar threshold was already used by other scientists, as the authors write in the conclusions.

The two thresholds of 80 and 95% were previously used by Wagner et al. [21] (80%), that analyzed 206 the climate drivers of the Amazon Basin forests greening, and by Tyukavina et al. [17] (83%), that 207 analyzed the types and rates of forest disturbance in Brazilian Legal Amazon between 2000 and 2013.

Finally, when the threshold is found, it is not discussed why it works and why did the others fail. This can be explained by knowing the trecover2000 methodology, but it is not explained here.

Some remarks about the paper

In the abstract: Please introduce the meaning of REDD+

Abstract: Knowing forest cover is an essential component of REDD+ policies,…”

Please explain the Treecover2000 method in detail.

It is very important to describe the data set. In this sense, please also explain the meaning of the abbreviations.

It is the 53 TreeCover2000 computed from a regression tree considering several metrics obtained from the 54 spectral bands Landsat (TOA reflectance and TOA brightness temperature and NDVI index) and 55 height of larger trees 5m obtained from the GLAS sensor.

Why a simple regression? What is the real reason proposed by Shimabokuro? Explain it, not just quote it.

Author Response

Responses to Reviewer 1 of the manuscript entitled:

Determining a Threshold to Delimit the Amazonian Forests from the Tree Canopy Cover 2000 GFC Data

By Gasparini K. A. C. and co-authors

We thank the Reviewer 3 for their comprehensive review and substantial comments. We appreciated the suggestions and comments. We also believe the results and conclusions of the paper are now more robust. Besides that, we performed an English language and style review of our manuscript.

Below, we repeat all Reviewer comments and reply to the concerns one by one. Each comment is numbered, using a continuous sequence and with our responses in bold.

1 - The paper introduces a method to estimate tree cover in the Amazon basin based on thresholding the available Treecover2000 data. The analysis is basically the comparison of thresholded results to a public reference data set.

The method leads to a valid result; nevertheless, the paper lacks scientific aspects.

R: We thank the Reviewer for their availability to review our manuscript.

2 - First, all data are previously available and the authors did not process them. Their work is a comparison by using thresholds between 0-100 to determine the best fit. In this step, the work is very naive and the tested values not justified. They just appear. Why not 82 instead of 80%?

R: The thresholds analysed in our manuscript were determined from preliminary tests and thresholds used in the literature. For example, in our tests, the 20% threshold forest cover did not differ from forest cover resulting from the 10% and 30% thresholds.

To address the reviewer' concern and clarify our text, we have inserted the following sentence into our "Methods" section: “We selected these thresholds based on preliminary tests and the previous studies from the literature [22,23,25–28], representing a wide range of thresholds.”

3 - It is not clear if the authors did a rigorous separation of the data set into test and control sets. It is recommended to use a set for finding the threshold and another one to verify if the method really works.

R: To address the reviewer's concern, we have modified part of our method to make it more robust. We modified the analysis to be able to evaluate the stability of independent data models from the bootstrap analysis. Please see our revised manuscript.

We now partition our entire study area into 9329 10 km by 10 km plots, where we calculate the proportion of forest cover for the threshold-maps and the reference map. Then, we performed a bootstrap approach with 10,000 interactions. Where, in each interaction, 10% of the 9329 plots were raffled with replacement. Then the means and standard deviations of the 10,000 R2, intercept, slope, and the RSE was calculated.

4 - The contribution of the paper is weak because a similar threshold was already used by other scientists, as the authors write in the conclusions.

The two thresholds of 80 and 95% were previously used by Wagner et al. [21] (80%), that analyzed 206 the climate drivers of the Amazon Basin forests greening, and by Tyukavina et al. [17] (83%), that 207 analyzed the types and rates of forest disturbance in Brazilian Legal Amazon between 2000 and 2013.

R: The authors disagree with the Reviewer. Although the authors cited used these thresholds, they were empirically chosen, as mentioned in our "Introduction" section. Thus, our study aimed to propose a threshold verified in a known and independent forest cover data set.

Besides, the sentence of our conclusion previously highlighted by the reviewer was erroneously inserted by us in “Conclusion” section. Our text is correct now.

5 - Finally, when the threshold is found, it is not discussed why it works and why did the others fail. This can be explained by knowing the trecover2000 methodology, but it is not explained here.

R: Please see our “Discussion and Conclusion” section.

6 - In the abstract: Please introduce the meaning of REDD+

“ Abstract: Knowing forest cover is an essential component of REDD+ policies,…”

R: The text has been corrected.

7 - Please explain the Treecover2000 method in detail.

It is very important to describe the data set. In this sense, please also explain the meaning of the abbreviations.

It is the 53 TreeCover2000 computed from a regression tree considering several metrics obtained from the 54 spectral bands Landsat (TOA reflectance and TOA brightness temperature and NDVI index) and 55 height of larger trees 5m obtained from the GLAS sensor.

R: Please see our revised “Introduction” section.

8 - Why a simple regression? What is the real reason proposed by Shimabokuro? Explain it, not just quote it.

R: The authors agree with the Reviewer. We have modified the following sentence from our "Methods” section to address the Reviewer' concern: “The proportions of forest cover within each of the 9329 plots defined by each threshold and the PRODES forest cover were compared using linear regression, suggested by Shimabukuro et al. [29]. This method is a robust and straightforward approach to validating remote sensing mapping.”

Reviewer 2 Report

Nice and useful paper!

Congratulations!

Author Response

Responses to Reviewer 2 of the manuscript entitled:

Determining a Threshold to Delimit the Amazonian Forests from the Tree Canopy Cover 2000 GFC Data

By Gasparini K. A. C. and co-authors

We are glad that the reviewer liked the narrative of the paper, its message, and the methodology used. We thank the reviewer for the comprehensive revision of the manuscript. Besides that, we performed an English language and style review of our manuscript.

Reviewer 3 Report

This study aimed to determine the best threshold for Tree Canopy Cover 2000 GFC product to make a forest mask for Amazon. The reference data for accuracy assessment consisted of a national forest cover map and 2550 Landsat pixels. Based on the results, a threshold of 80% is recommended for assessing forest cover in the Amazon.

The scope of the study is well-defined and rather narrow as expected for a letter. The study is limited to Mato Grosso but that is a rather large area and cover variation in tree cover. The manuscript is mostly well-written although there are some typos and clarifications needed. Figures are clear and well-prepared. In my consideration, this manuscript is publishable in Sensors after minor corrections (see details below).

L31. "second source", maybe "second largest source"?
L32. "which it is not known", not fluent
L39. Extra "]"
L44. Not clear what "repercussion" means here. Clarify or change the word.
L55. Remove "index" after NDVI, or give NDVI in full. "I" means index.
L55-56. "and height of larger trees 5m obtained from the GLAS sensor." This is misleading as GLAS was used only for validation. This sentence says it was used for input as well. Check Supplementary material of Hansen et al.
L59. Do not mix forest cover and canopy cover. I think TreeCover2000 gives canopy cover.
L92. What is the mapping scale of PRODES data? How is forest defined in this data? Clarify.
L134. "were cover by savannas formation vegetation [29]". Check this sentence. Add period.
L157. "samples plots" should be "sample plots"
L170. Period missing.
L199. How your results relate to tree cover threshold applied by Hansen et al. for forest change mapping? I think this should be discussed.

Author Response

Responses to Reviewer 3 of the manuscript entitled:

Determining a Threshold to Delimit the Amazonian Forests from the Tree Canopy Cover 2000 GFC Data

By Gasparini K. A. C. and co-authors

We thank the Reviewer 3 for their comprehensive review and substantial comments. We appreciated the suggestions and comments. We also believe the results and conclusions of the paper are now more robust. Besides that, we performed an English language and style review of our manuscript.

Below, we repeat all Reviewer comments and reply to the concerns one by one. Each comment is numbered, using a continuous sequence and with our responses in bold.

1 - This study aimed to determine the best threshold for Tree Canopy Cover 2000 GFC product to make a forest mask for Amazon. The reference data for accuracy assessment consisted of a national forest cover map and 2550 Landsat pixels. Based on the results, a threshold of 80% is recommended for assessing forest cover in the Amazon.

The scope of the study is well-defined and rather narrow as expected for a letter. The study is limited to Mato Grosso but that is a rather large area and cover variation in tree cover. The manuscript is mostly well-written although there are some typos and clarifications needed. Figures are clear and well-prepared. In my consideration, this manuscript is publishable in Sensors after minor corrections (see details below).

R: We are glad that the reviewer liked the narrative of the paper, its message, and the methodology used. We thank the reviewer for the comprehensive revision of the manuscript.

2 - L31. "second source", maybe "second largest source"?

R: The text has been corrected.

3 - L32. "which it is not known", not fluent

R: The text has been corrected.

4 - L39. Extra "]"

R: The text has been corrected.

5 - L44. Not clear what "repercussion" means here. Clarify or change the word.

R: The text has been corrected.

6 - L55. Remove "index" after NDVI, or give NDVI in full. "I" means index.

R: The text has been corrected.

7 - L55-56. "and height of larger trees 5m obtained from the GLAS sensor." This is misleading as GLAS was used only for validation. This sentence says it was used for input as well. Check Supplementary material of Hansen et al.

R: The text has been corrected.

8 - L59. Do not mix forest cover and canopy cover. I think TreeCover2000 gives canopy cover.

R: The text has been corrected.

9 - L92. What is the mapping scale of PRODES data? How is forest defined in this data? Clarify.

R: To address the reviewer's concern, we added the sentence "The data have a 1:250,000 mapping scale and represent areas of old-growth forests mapped manually from Radar and optical remote sensing data (http://terrabrasilis.dpi.inpe.br)." to our "Datasets" section.

10 - L134. "were cover by savannas formation vegetation [29]". Check this sentence. Add period.

R: The text has been corrected.

11 - L157. "samples plots" should be "sample plots"

R: The text has been corrected.

12 - L170. Period missing.

R: The text has been corrected.

13 - L199. How your results relate to tree cover threshold applied by Hansen et al. for forest change mapping? I think this should be discussed.

R: Thanks for this comment. Based on our findings, we find that less conservative thresholds eventually "create" forest cover beyond areas where they exist. Hansen et al. (2013) [1], for example, reported forest cover loss based on the 50% threshold for tree canopy cover, which includes loss in non-forest areas. These implications have been discussed extensively earlier in the literature [2,3].

In our “Discussion and Conclusion” section of manuscript we highlight: “The lower thresholds evaluated, between 10% and 50%, for example, overestimated forest cover, especially within savannah formations, where trees are sparser within the landscape. Thus, the choice of less conservative thresholds may inflate the quantification of forest disturbances, such as fires [26], deforestation [13], selective logging [22], and forest fragmentation [23–25] and consequently carbon emissions accounting.”

References

Hansen, M.C.; Potapov, P. V; Moore, R.; Hancher, M.; Turubanova, S. a; Tyukavina, A.; Thau, D.; Stehman, S. V; Goetz, S.J.; Loveland, T.R.; et al. High-Resolution Global Maps of 21st-Century Forest Cover Change. Science (80-. ). 2013, 342, 850–853. Hansen, M.; Potapov, P.; Margono, B.; Stehman, S.; Turubanova, S.; Tyukavina, A. Response to Comment on “High-resolution global maps of 21st-century forest cover change.” Science (80-. ). 2014, 344, 981–981. Tropek, R.; Sedla ek, O.; Beck, J.; Keil, P.; Musilova, Z.; Imova, I.; Storch, D. Comment on “High-resolution global maps of 21st-century forest cover change.” Science (80-. ). 2014, 344, 981–981.

Round 2

Reviewer 1 Report

The term residual standard error (RSE) is not well known in the community, but is familiar in "R" software.
Please consider using RMSE (root mean squared error) instead.

Please give information about the method, not the command name.
Readers don't know this functions; what do they do? what is the input? what is the method? what is the output?

132 To calculate the forest proportion within each
133 of the 10 by 10 km plots, the "lsm_c_pland" function of the "landscapemetrics" package was used.

equation 1-2.
You use "*" for multiplication in (1) and not in (2); please keep a standard. The best option, in my opinion, is the use of a dot.

199 The “Accuracy Assessment” tool results showed that poorest match
again, don't refer to "tools", describe the method.
Please think about: How is it supposed that someone that does not use "R" can replicate the experiments and analysis?

The paper was improved. Congratulations.

Author Response

Responses to Reviewer 1 of the manuscript entitled:

Determining a Threshold to Delimit the Amazonian Forests from the Tree Canopy Cover 2000 GFC Data

By Gasparini K. A. C. and co-authors

The authors thank Reviewer 1 for their comments and suggestions that have improved and made our manuscript more robust.

Below, we repeat all the Reviewer' comments and reply to them one by one. Each comment is numbered, using a continuous sequence and with our responses in bold.

1 - The term residual standard error (RSE) is not well known in the community, but is familiar in "R" software. Please consider using RMSE (root mean squared error) instead.

R: We modified the text according to the Reviewer’s suggestion.

2 - Please give information about the method, not the command name.

Readers don't know this functions; what do they do? what is the input? what is the method? what is the output?

132 To calculate the forest proportion within each

133 of the 10 by 10 km plots, the "lsm_c_pland" function of the "landscapemetrics" package was used.

equation 1-2.

R: In the first two paragraphs of our "Methods" section, we detail the whole approach taken in our manuscript. From this description, we believe that the readers can replicate our method on different platforms, not just in the R software.

To give recognition to the developers of the R software and the packages used, we wrote the last part of the second paragraph of our "Methods" section. With the same goal, we wrote the last paragraph of our "Methods" section.

3 - You use "*" for multiplication in (1) and not in (2); please keep a standard. The best option, in my opinion, is the use of a dot.

R: We modified the equations according to the Reviewer’s suggestion.

4 - 199 The “Accuracy Assessment” tool results showed that poorest match

again, don't refer to "tools", describe the method.

R: We modified the text to address the Reviewer’s concern.

5 - Please think about: How is it supposed that someone that does not use "R" can replicate the experiments and analysis?

R: Please see our response to the second comment.